# FORGEDIT: TEXT GUIDED IMAGE EDITING WITH VIA LEARNING AND FORGETTING

## ABSTRACT

Text guided image editing is a recently popular but challenging task. It requires an editing model to estimate by itself which part of the image should be edited, and then perform complicated non-rigid editing while preserving the characteristics of original image. Previous fine-tuning based approaches are often time-consuming and vulnerable to overfitting, which catastrophically limits their editing capabilities. To tackle these issues, we design a novel text guided image editing method, named as Forgedit. First, we propose a novel fine-tuning framework able to reconstruct a given image efficiently by jointly learning vision and language information. Then we introduce vector subtraction and projection mechanisms to explore accurate text embeddings for editing. We also find a general property of UNet structures in Diffusion Models, which inspired us to design new forgetting strategies to diminish the fatal overfitting issues, significantly boosting the editing abilities of Diffusion Models. Our method, Forgedit, built on Stable Diffusion, achieves new state-of-the-art results on the challenging text guided image editing benchmark: TEdBench, surpassing the previous SOTA methods such as Imagic (even built on stronger Imagen), in terms of both CLIP score and LPIPS score.

## 1 INTRODUCTION

Image Editing (Oh et al., 2001) is a fundamental problem in computer vision. To perform image editing, a guidance information or condition is often provided to the model to indicate the editing intention. Text information is the most direct and general form of such editing guidance, in which case the editing task is called text guided image editing. Such a text describing the content of the desired edited image is usually called target prompt. In this paper, we aim to tackle the task of text guided image editing, with only an original image and a target prompt provided - which are the minimum requirements of inputs for text guided image editing. Text guided image editing contains both rigid and non-rigid editing, such as, editing the appearance, identity and style, replacing or removing certain parts of the image, editing the pose, action and angles of the objects, editing multiple objects with complex relationships, controlling the numbers and positions of the objects, etc. According to whether fine-tuning process is performed, the approaches of text guided image editing are generally categorized into optimization-based methods and non-optimization ones.

Recently, a number of non-optimization editing methods have been developed, such as Control-Nets(Zhang & Agrawala, 2023), diffusion based inpainting models (Rombach et al.; Avrahami et al., 2022), SDEdit (Meng et al., 2021), PnP Diffusion (Tumanyan et al., 2023), Instruct Pix2pix (Brooks et al., 2023), DiffEdit (Couairon et al., 2023). These methods have made significant improvements on non-optimization editing, but existing approaches still suffer from many difficulties on preserving the characteristics of original image, or performing sophisticated and accurate non-rigid edits. Thus, it is critical to fine-tune a diffusion model with the original image, enabling the model to learn and encode more detailed information of the identity and characteristics. For example, Imagic (Kawar et al., 2023) is a three-stage text guided image editing method, which regards the target prompt as a pseudo source prompt that describes the original image. Built on the powerful Imagen (Saharia et al., 2022), Imagic is the current state-of-the-art text guided image editing algorithm. However, such a multi-stage fine-tuning process takes long time with great computational cost. Furthermore, recent DreamBooth (Ruiz et al., 2023) is a popular and powerful fine-tuning method for learning new concepts with few examples. We adapted and improved DreamBooth to perform text guided image editing. Instead of requiring a user provided prompt 'a [V] object' to refer to the editing

object, we utilize BLIP (Li et al., 2022) to generate a caption describing the original image. We call such an editing method BLIP+DreamBooth, elaborated in appendix A.1. Such BLIP+DreamBooth combinations are capable of conducting non-rigid edits while preserving more consistent characteristics of original image, demonstrating excellent semantic alignments with the target prompt, and high fidelity to the original image. However, Both Imagic and BLIP+DreamBooth may suffer from overfitting problems, weakening the editing capability of current diffusion models.

In this paper, we are going to tackle the aforementioned issues of optimization based editing methods. We name our text guided image editing method as *Forgedit* (similar to *forget it*), which consists of two stages: fine-tuning and editing. Overall, with a generated source prompt from BLIP (Li et al., 2022), we design a new vision and language joint optimization framework, which allows the model to reconstruct the original image using the source text embedding and the UNet in a particularly efficient manner. For instance, the fine-tuning process can be accomplished in less than one minute on a single A100 GPU, which is much faster than Imagic (Kawar et al., 2023) built on Stable Diffusion (Rombach et al.), which would take about 7 minutes on an A100 GPU.

Besides, we explore two different methods to merge the source and the target text embeddings: vector subtraction and projection. Vector subtraction computes a sum of source prompt embedding and a weighted subtraction of the target prompt embedding and source prompt embedding. Instead, vector projection decomposes the target prompt embedding along source prompt embedding and its orthogonal direction, and then sum the two decomposed vectors with two coefficients. We found that vector subtraction is able to improve editing performance, while vector projection allows for preserving more characteristics of original image during editing.

Finally, our Forgedit is designed to overcome the common overfitting issue existing in current optimization-based editing methods. The overfitting issue easily leads the image editing process towards the reconstruction of the original image instead of the editing target. Straightforward solutions may be trying different learning rates, training steps, or selecting proper parameters of diffusion models to fine-tune. Yet, there are no silver bullets to find a group of proper hyper-parameters for each editing image, thus such hyper-parameter searching for the fine-tuning process could be highly inefficient and resource consuming. Instead, we propose a new *forgetting strategy* to tackle the overfitting issue during *sampling* process. Compared to fine-tuning process, the sampling process is more computational efficient. Such a forgetting strategy is carefully designed based on our observation of an important property of UNet structures in diffusion models: the encoder part of UNets mainly focuses on learning information of pose, action, angles, and spatial positions, meanwhile the decoder part of UNets is capable of modelling details of appearance and textures. This inspired us to replace the learned parameters of UNets with the original parameters according to the purpose of target prompt, which we call *forgetting*.

To sum up, our main contributions are:
1. We present Forgedit, a novel optimization based image editing framework, capable of performing both rigid and non-rigid editing, while effectively tackling multiple general challenges on this task.
2. We introduce new vector projection mechanism which provides an more controllable alternative for computing the combination of the source text embedding and the target text embedding. This improves Forgedit's capability for preserving more consistent characteristics of original image than existing methods based on vector subtraction.
3. We design a novel forgetting strategy based on our observation on the UNet architecture of diffusion models. This allows us to effectively tackle the critical overfitting issue on optimization based image editing methods, which thus significantly boosts the editing capability of diffusion models.

Our Forgedit can achieve new state-of-the-art results on the challenging benchmark TEdBench (Kawar et al., 2023) (even by using an outdated Stable Diffusion 1.4), surpassing previous SOTA Imagic built on Imagen in terms of both CLIP score (Hessel et al., 2021) and LPIPS score (Zhang et al., 2018). Our Forgedit is highly flexible, and is readily applicable to other fine-tuning based text guided image editing methods, with significant performance improvements. We will show more applications for such extensions in the appendix A.2.

Due to limited pages, we have to move details of related works to appendix A.3.

## 2 FORGEDIT

**Problem settings.** Given a target prompt and an image, text guided image editing is performed according to the provided target prompt. It not only requires the editing being well-conducted by matching target prompt accurately, but also asks to preserve the detailed characteristics of original image as much as possible. In this work, we attempt to tackle text guided image editing with a condition that only a target prompt and an original image are provided, which means that the model should estimate by itself which region of the given image is relevant to the target prompt, and then conduct image edit accurately on this region. We aim to design a general editing method, which is capable to perform different kinds of edits including both rigid and non-rigid editing.

### 2.1 PRELIMINARIES

Diffusion models (Ho et al., 2020; Sohl-Dickstein et al., 2015) consist of a forward process and a reverse process. The forward process starts from the given image $x_0$, and then progressively add Gaussian Noise $\epsilon_t \sim \mathcal{N}(0,1)$ in each timestep $t$ to get $x_t$. In such a diffusion process, $x_t$ can be directly calculated at each timestep $t \in \{0, ..., T\}$,

$$x_t = \sqrt{\alpha_t}x_0 + \sqrt{1-\alpha_t}\epsilon_t \tag{1}$$

with $\alpha_t$ being diffusion schedule parameters with $0 = \alpha_T < \alpha_{T-1}... < \alpha_1 < \alpha_0 = 1$ .

In the reverse process, given $x_t$ and text embedding $e$, the time-conditional UNets $\epsilon_\theta(x_t, t, e)$ of diffusion models predict random noise $\epsilon_t$ added to $x_{t-1}$. With DDIM (Song et al., 2021), the reverse process can be computed as,

$$x_{t-1} = \frac{\sqrt{\alpha_{t-1}}}{\sqrt{\alpha_t}}(x_t - \sqrt{1-\alpha_t}\epsilon_\theta(x_t, t, e)) + \sqrt{1-\alpha_{t-1}}\epsilon_\theta(x_t, t, e) \tag{2}$$

With Latent Diffusion Models (Rombach et al.), the original image $x_0$ is replaced by a latent representation $z_0$ obtained from a VAE (Kingma & Welling, 2014) Encoder $\varepsilon(x_0)$. The overall training loss is computed as,

$$L = \mathbb{E}_{z_t, \epsilon_t, t, e}||\epsilon_t - \epsilon_\theta(z_t, t, e)||_2^2 \tag{3}$$

### 2.2 JOINT FINE-TUNING

In order to tackle such challenging text guided image editing problems, we have to fine-tune the model to learn the concepts from the original image thus the model could reconstruct them consistently during the editing process. It is worth noting that although DDIM inversion (Song et al., 2021) could reconstruct the original image, the given text prompt has to be an empty string. If the given text prompt is not empty, DDIM inversion is incapable to reconstruct original image precisely and often leads to significant appearance shift (Hertz et al., 2023; Meng et al., 2021). Thus it is necessary to optimize the network for high quality reconstruction and semantic understanding. As shown in Figure 1, we introduce the overall design of our vision-language joint optimization framework.

**Source prompt generation.** We first use BLIP (Li et al., 2022) to generate a caption describing the original image, which is referred to as the source prompt. The source prompt is then fed to the text encoder of Stable Diffusion (Rombach et al.), generating an embedding $e_{src}$ of source prompt. Previous three-stage editing method Imagic (Kawar et al., 2023) regards target prompt text embedding as source one $e_{src}$. We found that it is essential to use the BLIP caption instead of using the target prompt as a pseudo source prompt like Imagic. Otherwise such fine-tuning methods easily lead to overfitting issues, as demonstrated in the 5th column of Figure 7.

**Vision-language joint learning.** We choose to optimize encoder layers of 0, 1, 2 and decoder layers of 1, 2, 3 in the UNet structure since we found that fine-tuning deepest features would lead to overfitting in our Forgedit framework. Similar with Imagic, we regard source text embedding as parameters of the network. Yet different with Imagic, we found it vital to jointly optimize the source text embedding and UNet parameters, which is of great importance for faster learning and better reconstruction quality. In particular, due to a large domain gap between text and image, we

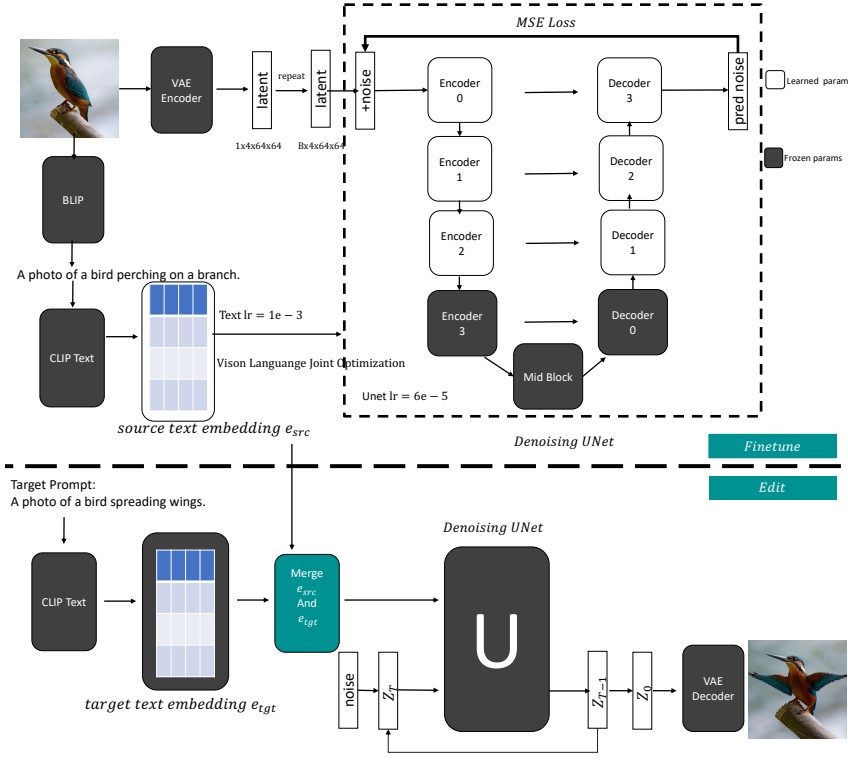

Figure 1: Overall framework of our Forgedit, consisting of a joint fine-tuning stage and an editing stage. We use BLIP to generate a text description of an original image, and compute an embedding of the source text $e_{src}$ using a CLIP text encoder. The source embedding $e_{src}$ is then jointly optimized with UNet using different learning rates for text embedding and UNet, where the deep layers of UNet are frozen. During the editing process, we merge the source embedding $e_{src}$ and the target embedding $e_{tgt}$ with vector subtraction or projection to get a final text embedding $e$. With our forgetting strategies applied to UNet, we utilize DDIM sampling to get the final edited image.

use different learning rates for source text embedding ($10^{-3}$) and UNet ($6 \times 10^{-5}$), with an Adam Optimizer (Kingma & Ba, 2015). For faster training, since we only have a single training image, we repeat the tensors on batch dimension for batch-wise optimization with a batch size of 10. We use mean square error loss, and empirically found that stable reconstruction results can be achieved when the final loss is less than 0.03. With a batch size set to 10, the models are fine-tuned for 35 to 40 steps. We stop the training over 35 steps when the loss is less than 0.03, or stop at 40 steps at most. This fine-tuning process is significantly more efficient than Imagic, by taking less than 1 minute on a single A100 GPU. The training loss is computed as,

$$L = \mathbb{E}_{z_t, \epsilon_t, t, e_{src}} ||\epsilon_t - \epsilon_{\theta, e_{src}}(z_t, t, e_{src})||_2^2 \tag{4}$$

where the main difference with the training loss presented in 3 is that $e_{src}$ is considered as parameters to optimize. Our joint optimization brings the model the strong capability to reconstruct the original image, given the optimized source text embedding $e_{src}$.

## 2.3 REASONING AND EDITING

We first input the target prompt to the CLIP (Radford et al., 2021) text encoder of the Stable Diffusion model (Rombach et al.), computing a target text embedding $e_{tgt}$. With our learned source text embedding $e_{src}$, we propose two methods to compute a combination of $e_{src}$ and $e_{tgt}$ so that the combined text embedding can be applied for editing the original image according to the target prompt. Given $e_{src} \in \mathbb{R}^{B \times N \times C}$ and $e_{tgt} \in \mathbb{R}^{B \times N \times C}$, we conduct all vector operations on the $C$ dimension to get the final text embedding $e$.

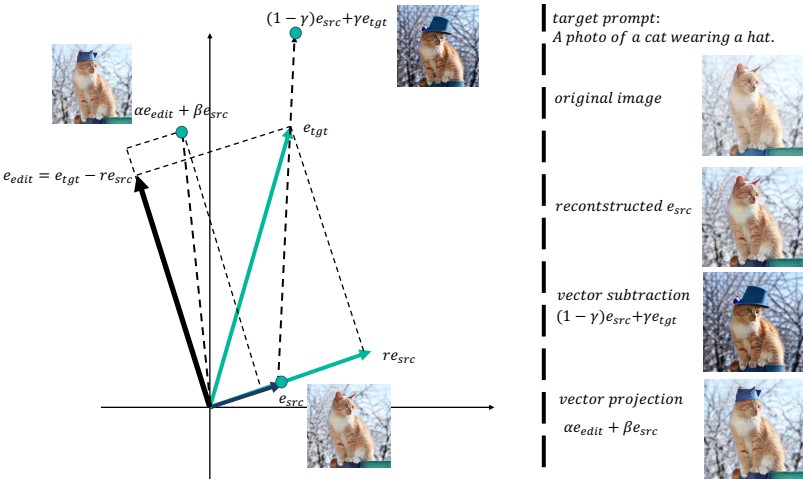

Figure 2: We demonstrate vector subtraction and vector projection to merge $e_{src}$ and $e_{tgt}$. Vector subtraction could lead to inconsistent appearance of the object being edited since it cannot directly control the importance of $e_{src}$. The vector projection decomposes the $e_{tgt}$ into $re_{src}$ along $e_{src}$ and $e_{edit}$ orthogonal to $e_{src}$. We can directly control the scales of $e_{src}$ and $e_{edit}$ by summation.

**Vector Subtraction.** We use the same interpolation method as Imagic (Kawar et al., 2023),

$$e = \gamma e_{tgt} + (1 - \gamma)e_{src} = e_{src} + \gamma(e_{tgt} - e_{src}) \tag{5}$$

As shown in Figure 2, the final text embedding $e$ is obtained by travelling along vector subtraction $e_{tgt} - e_{src}$ . In our experiments, we found that in most cases, $\gamma$ goes beyond 1 when the editing is performed successfully. This leads to a problem that the distance between the final embedding $e$ and the source embedding $e_{src}$ may be so far that the appearance of the edited object could change vastly.

**Vector Projection.** We propose to use vector projection to better preserve the appearance of the original image. As shown in the Figure 2, we decompose a target prompt text embedding $e_{tgt}$ into a vector along $e_{src}$ and a vector orthogonal to $e_{src}$. We call the orthogonal vector $e_{edit}$. We first calculate the ratio $r$ of the projected vector on $e_{src}$ direction.

$$r = \frac{e_{src}e_{tgt}}{||e_{src}||^2} \tag{6}$$

Thus, we could get the $e_{edit}$ by computing

$$e_{edit} = e_{tgt} - re_{src} \tag{7}$$

To preserve more details of original image, we sum $e_{src}$ and $e_{edit}$ with coefficient $\alpha$ and $\beta$,

$$e = \alpha e_{src} + \beta e_{edit} \tag{8}$$

**Editing.** We use DDIM sampling (Song et al., 2021) with a classifier free guidance (Ho, 2022) to conduct the edit. The guidance scale is 7.5. For vector subtraction, we iterate over a range of $\gamma \in [0.8, 1.6]$. For vector projection, we choose $\alpha$ from two values $\{0.8, 1.1\}$, and $\beta$ from a range of $[1.0,1.5]$.

## 2.4 FORGETTING STRATEGY

**Forgetting mechanism** In some cases the network still overfits since there is only one training image provided. The fine-tuning process is computational expensive compared to sampling process, thus we design a forgetting strategy during sampling process to tackle the overfitting problem. The network is only fine-tuned once, and can be converted to multiple different networks during sampling process by merging certain fine-tuned parameters $w_{learned}$ and the corresponding parameters of

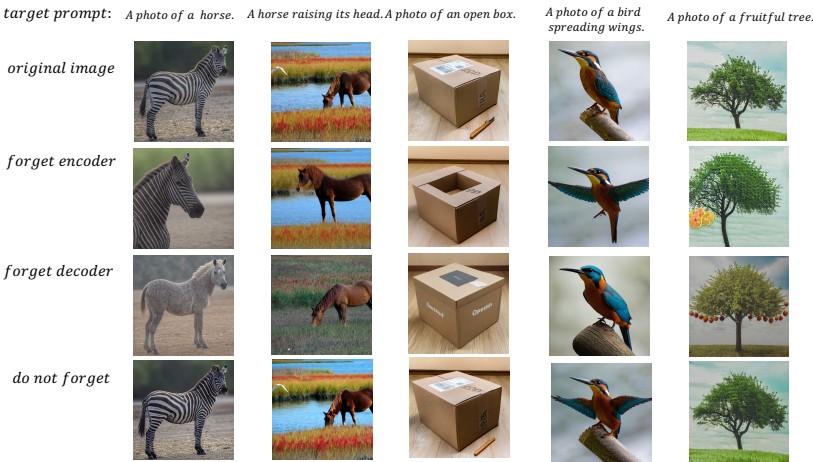

Figure 3: The encoder of UNets learn features related to pose, angle, structure and position. The decoder are related to appearance and texture. Thus we design a forgetting strategy according to the editing target.

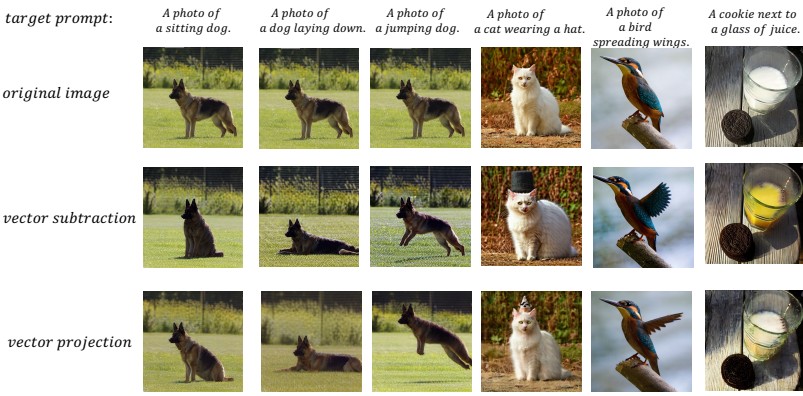

Figure 4: Comparisons of vector subtraction and vector projection, which are complementary.

original UNet (before fine-tuning) $w_{orig}$, with a balance coefficient $\sigma$. In practice, we found that $\sigma = 0$ works in general, which means that we can simply replace the fine-tuned parameters with original parameters so that the network completely forgets these learned parameters.

$$w = \sigma w_{learned} + (1 - \sigma)w_{orig} \tag{9}$$

**UNet's property** As shown in Figure 3, we found an interesting property of UNets in diffusion models. The encoder of UNets learns space and structure information like the pose, action, position, angle and overall layout of the image, while the decoder learns appearance and identity instead.

**Forgetting strategy** If the target prompt tends to edit space information, for example, the pose or layout, we choose to forget parameters of the encoder. If the target prompt aims to edit the appearance, the parameters of decoder should be forgotten. Currently we only apply the forgetting strategy when a text embedding $e$ is obtained by vector subtraction in previous section. For editing with the forgetting strategy, we iterate over a range of $\gamma \in [0.0, 1.4]$. For different settings of forgetting strategy, we explore their effects in ablation study, as shown in Figure 5 and Figure 6.

**Limitations.** Our Forgedit has a number of limitations which are elaborated in appendix A.6.

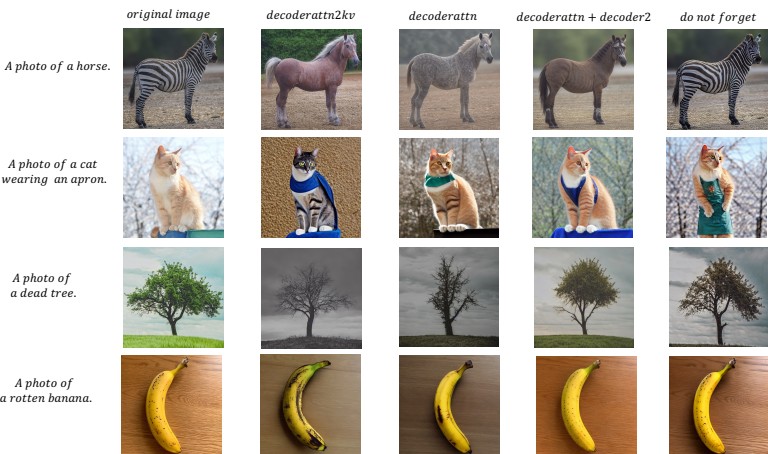

Figure 5: We explore various forgetting strategies for decoder. All learned encoder parameters are preserved. In the $2^{nd}$ to $4^{th}$ columns, we preserve decoder cross-attention parameters, decoder self-attention and cross-attention, decoder self-attention, cross-attention and the entire decoder2 block, forgetting all the other parameters of decoder.

## 3 EXPERIMENTS

**Benchmark.** TEdBench (Kawar et al., 2023) is one of the most difficult public-available text guided image editing benchmarks. It contains 100 editings, with one target prompt and one image for each edit. These target prompts are very general with diversity, including but not limited to changing the appearance of objects, replacing certain parts of the image, changing the position, action and number of the object, editing multiple objects with complex interactions. In particular, non-rigid edits turn out to be very tough for many SOTA text-guided image editing methods. In terms of quantitative evaluation, we utilize CLIP Score (Hessel et al., 2021) to measure semantic alignments with target prompt, and LPIPS score (Zhang et al., 2018) to indicate fidelity to the original image.

### 3.1 ABLATION STUDY

**Vector subtraction vs vector projection.** We compare two different reasoning methods to merge $e_{src}$ and $e_{tgt}$ to get the final text embedding $e$, shown in Figure 4 . For the dog and the cat examples, vector projection can preserve more details of the appearance of the dog and the cat than vector subtraction. However, for a glass of milk and cookie example, vector subtraction performs better than vector projection which struggles to change the milk to juice and also introduces wave-like blurs in the image. We observe such phenomenons in many other cases for vector projection, which demonstrates that it is more suitable for edits where the identity of object should be kept instead of changed. These two methods are complementary to each other, with vector projection better at preserving the identity, and vector subtraction better at editing.

**Forgetting strategy.** Our forgetting strategy enhances editing ability of our model, but forgetting parameters would inevitably lead to minor reconstruction quality. For example, as shown in Figure 3, for an encoder or a decoder, we keep all parameters related to self-attention and cross-attention, while forgetting the rest, which are called 'encoderattn' in Figure 6 and 'decoderattn' in Figure 5. We found that there are certain unexpected changes unrelated to the target prompt, which are the side effects of forgetting strategy. For each column, the background of image changes slightly, the white bird disappears, the knife is gone, the branch no longer exists, the appearance of the tree changes.

We conduct experiments with different extent of forgetting strategies. In Figure 5 , we explore different decoder forgetting strategies. With all fine-tuned parameters of encoder preserved and all decoder parameters forgotten, we gradually add fine-tuned parameters back to decoder. 'decoderattn2kv' means that we use fine-tuned parameters of decoder cross-attention key and value matrices. Since all the fine-tuned encoder parameters are preserved, the overall structure of the image and the pose of the objects being edited are almost identical with the original image, yet the appearance and textures are changed. 'decoderattn' indicates that we utilize all learned self-attentions and cross-attentions parameters in the decoder. This is our default setting since it is general. More appearance

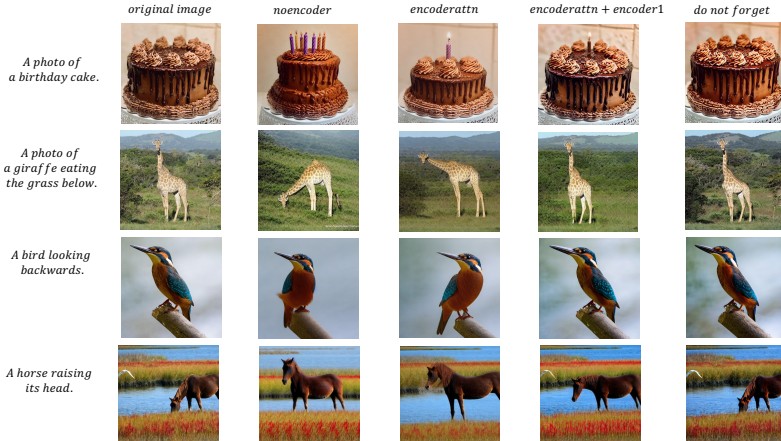

Figure 6: We explore different forgetting strategies for encoder. All learned decoder parameters are preserved. For the second to fourth column each, we preserve none of the encoder parameters, encoder self attention and cross attention, encoder self attention and cross attention and the entire encoder1 block, forgetting all the other parameters of encoder.

and textures features of the original image are preserved in such a setting. 'decoderattn+decoder2' refers to the forgetting strategy that we preserve all learned self-attentions and cross-attentions of decoder plus the decoder2 block. The position of decoder2 block is shown in Figure 1. More details are preserved for some edits, yet for the others the editing ability of our method is lost due to overfitting. In the last column of figure, we show editing results by using all fine-tuned parameters.

We explore different forgetting strategies for an encoder in Figure 6. 'noencoder' indicates that we forget all learned parameters of the encoder and only use the learned decoder parameters for sampling. 'encoderattn' refers to a strategy that we preserve all the parameters of self-attention and cross-attention. With 'encoderattn+encoder1' strategy, we preserve encoder self-attention, cross-attention and the encoder1 block. All the other parameters of encoder are forgotten.

## 3.2 COMPARISON WITH STATE-OF-THE-ART

We compare our Forgedit with SOTA methods in Figure 7. For non-optimization text guided image editing methods, we choose to compare with the most representative method, SDEdit (Meng et al., 2021), which we found struggles to preserve the identity of the edited objects in most cases.

We also compare with a kind of strong optimization involved method, which we call 'BLIP+DreamBooth', shown in the 3rd and 4th column of Figure 7. For details of this BLIP+DreamBooth method we created for text guided image editing, please refer to appendix A.1 due to page limits. It is obvious that BLIP+DreamBooth with Text Encoder and UNet jointly optimized is much better at preserving the identities and backgrounds than SDEdit. However, BLIP+DreamBooth is still prune to overfitting. We found that Forgedit can also alleviate the overfitting of BLIP+DreamBooth, which is introduced in appendix A.2.

We also compare with the SOTA three-stage text guided image editing method, Imagic (Kawar et al., 2023). We use Stable Diffusion (Rombach et al.) and Imagen Saharia et al. (2022) as the Diffusion Models for Imagic respectively, shown in the 5th and 6th columns of Figure 7. Imagic with Stable Diffusion suffers greatly from overfitting, leading to few successful edits. Imagic with Imagen is the current SOTA on TEdBench, demonstrating very strong editing abilities and preserves the original identities well in most cases. Our method, Forgedit, shown in the last column, though with the inferior Stable Diffusion as diffusion models for editing, is generally on par with Imagic with Imagen in most cases, sometimes better. Also, our Forgedit with the outdated Stable Diffusion 1.4 surpasses the current SOTA Imagic+Imagen on TEdBench benchmark in terms of both CLIP Score and LPIPS Score, shown in Table 1.

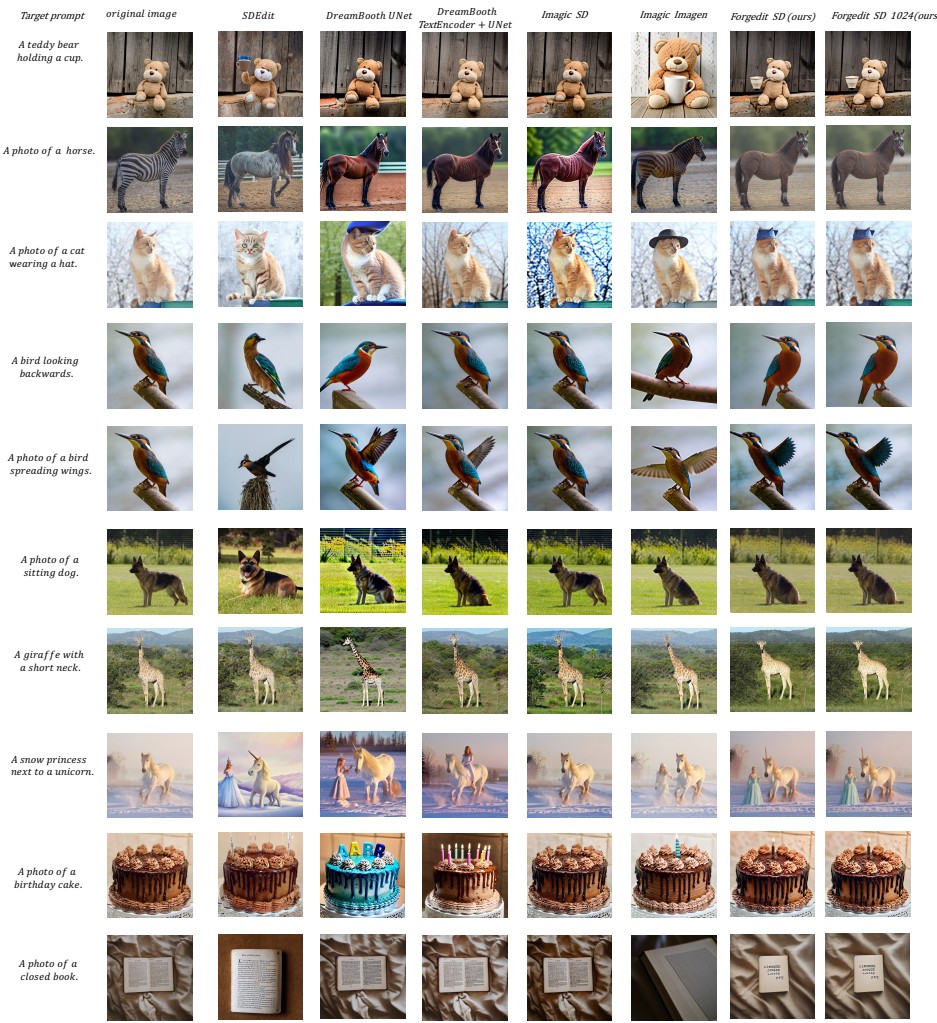

Figure 7: Comparison with SOTA: non-optimization SDEdit, optimization BLIP+DreamBooth and Imagic, demonstrating the strong editing ability and stable identity preservation.

| Editing method | CLIP Score ↑ | LPIPS Score ↓ | FID Score ↓ |
|---|---|---|---|
| Imagic+Imagen (Kawar et al., 2023) | 0.748 | 0.537 | 8.353 |
| Forgedit+SD (ours) | **0.771** | **0.534** | **7.071** |

Table 1: Our Forgedit with Stable Diffusion is the new state-of-the-art text guided image editing method on the challenging benchmark TEdBench, surpassing previous SOTA Imagic+Imagen.

## 3.3 CONCLUSION

We present our Forgedit framework to tackle the challenging text guided image editing problem. Besides the optimized vision language joint learning for fast reconstruction of the original image, we also introduce new vector projection mechanism to enhance the capability of identity preservation during editing. Finally, we propose a forgetting strategy to effectively solve the overfitting issue of optimization based model during sampling. Even with the outdated Stable Diffusion 1.4, our Forgedit achieves new SOTA CLIP score and LPIPS score on challenging editing benchmark TEdBench. Forgedit can be readily applicable to other fine-tuning based methods like BLIP+DreamBooth. We demonstrate generalization of Forgedit in the appendix. Theoretically, our Forgedit framework should also be compatible with other structures of diffusion models beyond SD thus has the potential to obtain better editing results.

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

## A APPENDIX

### A.1 BLIP+DREAMBOOTH

In order for DreamBooth (Ruiz et al., 2023) to be applied in text guided image editing, we utilize BLIP (Li et al., 2022) to generate captions describing the original image like our Forgedit. With the caption, we train the UNet to reconstruct the original image and edit the image by directly using the target prompt to guide the fine-tuned UNet for image generation, shown in the 3rd column of Figure 7. We also experiment with an improved version by training text encoder and UNet at the same time, shown in the 4th column. Such simple fine-tuning of UNet and text encoder are actually very powerful text guided image editing methods. Our BLIP+DreamBooth uses BLIP generated caption yet original DreamBooth requires user provided caption in a special form of 'a [V] object' referring the object to be reconstruct. Following the settings of DreamBooth (Ruiz et al., 2023), we use a learning rate of $5 \times 10^{-6}$ for both text encoder and UNet, with a batch size of 4. We train BLIP+DreamBooth with one image for 100 steps, which takes more than one minute on a A100 GPU. Unlike original DreamBooth which needs 3 to 4 images to learn the new object concept, we found that with BLIP+DreamBooth one image is enough to reconstruct the majority features of the original image. However, BLIP+DreamBooth, when only UNet is fine-tuned, suffers from underfitting since it cannot preserve the identity of the objects in many cases. BLIP+DreamBooth suffers from overfitting in many cases when text encoder and UNet are jointly fine-tuned. In fact, we found that our Forgedit can also be simply adapted to help tackling such overfitting issues of BLIP+DreamBooth, shown in the A.2, which again demonstrates the strong generalization of Forgedit framework on various optimization based editing methods.

## A.2 DREAMBOOTH+FORGEDIT

Forgedit is a very general framework, whose main features come from three aspects: joint vision and language learning with original image, obtaining final text embedding by vector subtraction and vector projection, using forgetting strategies to tackle the overfitting issues. Here we show how to extend Forgedit to BLIP+DreamBooth (Li et al., 2022; Ruiz et al., 2023). It is also possible to adapt our Forgedit to other Diffusion Models (Ho et al., 2021; Saharia et al., 2022) or fine-tuning methods (Hu et al., 2022), which we will explore in the future.

**vision and language joint learning** This is natural for the method which we call BLIP+DreamBooth Text Encoder and UNet, since the Text Encoder and UNet are jointly trained already.

**vector subtraction and vector projection** Our Forgedit presented in the main paper regards the source text embedding as a part of the network to optimize. For BLIP+DreamBooth, since we have already fine-tuned the text encoder, we switch to use the text encoder to get source text embedding directly from source prompt. Now we can use vector subtraction and vector projection in the same way.

**forgetting strategy** We could directly apply the forgetting strategies to BLIP+DreamBooth. However, since the information are injected into both text encoder and UNet, our forgetting strategies on UNet may still fail in some cases. We will explore the forgetting strategies in text encoder in the future.

We show some cases in Figure 8, and compare the editing effects of BLIP+DreamBooth+Forgedit with previous state-of-the-art text guided image editing methods and our Forgedit presented in the main paper. Comparing the 4th column and the last column, we could find that with Forgedit framework, the editing ability of DreamBooth has been tremendously improved. Please note that DreamBooth+Forgedit is not a simple combination of our Forgedit presented in the main paper and BLIP+DreamBooth, since the fine-tuning process of our Forgedit is different with Dream-Booth+Forgedit. This leads to the fact that DreamBooth+Forgedit is not always better than Forgedit, shown in the last two columns in Figure 8.

## A.3 RELATED WORK

Due to limited pages, we have no choice yet have to move the related works to appendix.

**Text to Image Diffusion Models** Diffusion Models have dominated text to image generation. DDPM(Ho et al., 2020) improves Diffusion process proposed by Sohl-Dickstein et al. (2015) on generating images. DDIM (Song et al., 2021) accelerates the sampling procedure of Diffusion Models by making reverse process deterministic and using sub-sequence of time-steps. Dalle 2 (Ramesh et al., 2022) trains a diffusion prior to convert a text caption to CLIP (Radford et al., 2021) image embedding and then employs a Diffusion Decoder to transfer the generated CLIP image embedding to an image. Imagen (Saharia et al., 2022) is a Cascaded Diffusion Model (Ho et al., 2021), whose UNet is composed of three Diffusion Models generating images with increasing resolutions. Also, Imagen employs the powerful T5 text encoder (Raffel et al., 2020), which turns out to be vital for complex semantic understanding and generating sophisticated scenarios. Stable Diffusion (Rombach et al.) utilizes Variational AutoEncoders (Kingma & Welling, 2014) to compress the training image to a compact latent space so that the UNets could be trained with low resolution latents in order to save computational resources.

**Image Editing with Diffusion Models** Empowered by recent progress in text-to-image Diffusion Models, image editing methods have witnessed remarkable improvements. There are various works for non-optimization editing. ControlNets (Zhang & Agrawala, 2023) are trained on extra datasets to learn generating images with different conditions. However, these conditions only reflect partial attributes of the original image thus ControlNets are incapable of preserving the identity of the object being edited and also struggle to conduct non-rigid edits. Inpainting Models based on Diffusion Models(Rombach et al.; Avrahami et al., 2022) require masks to indicate the editing region, for whom the target mask can be obtained via semantic segmentation models by using a text prompt to refer to. Such text guided Inpainting Models are good at replacing or removing objects, better than other text guided image editing models in terms of preserving non-edited details of original image. However, there are several disadvantages of text guided inpainting models. First, these models cannot preserve the identity of the object being edited. Second, due to the restricts of the region of

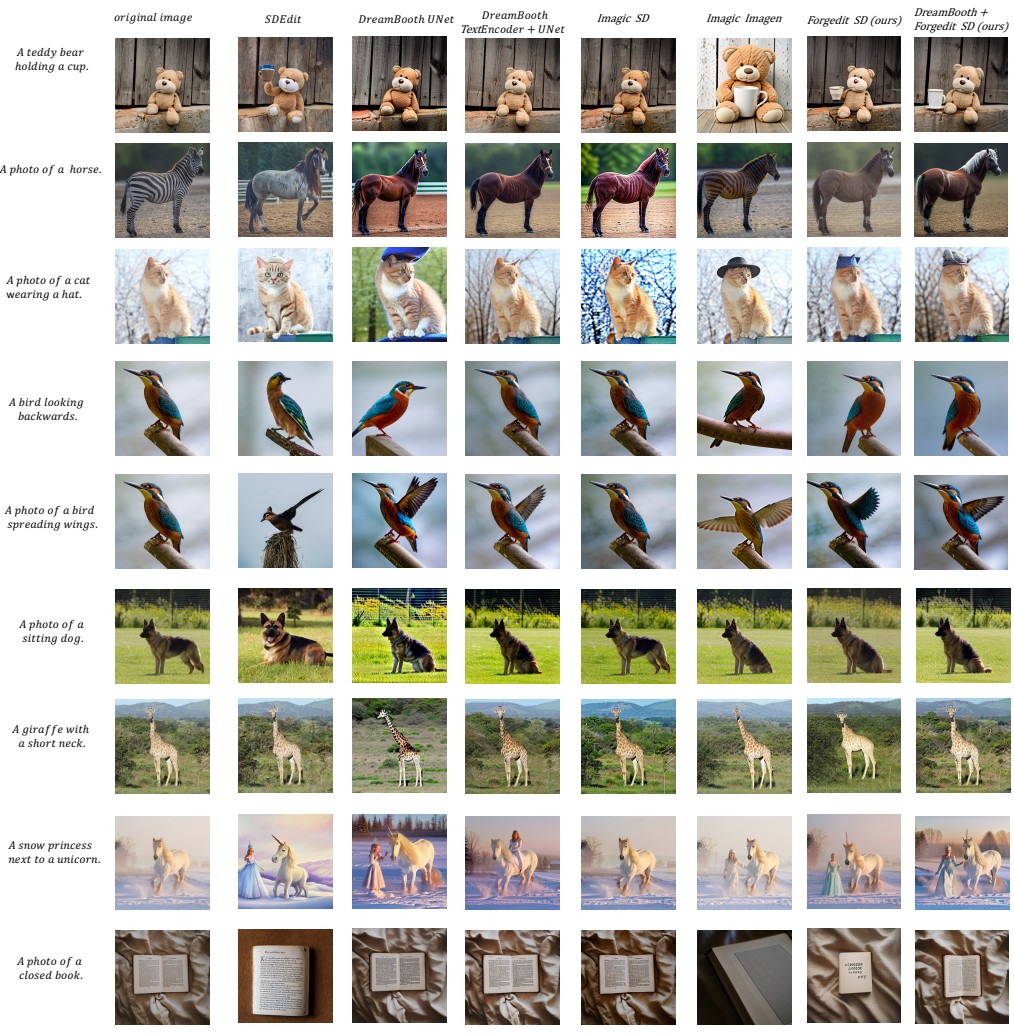

Figure 8: The editing effects of DreamBooth+Forgedit and comparisons with SOTA methods. Please note that DreamBooth+Forgedit is not always better than Forgedit, as shown in the last two columns.

masks, inpainting models cannot conduct non-rigid editing, for example, making a bird perching on the branch spread its wings. Third, extra masks or texts to refer to the target objects in original image has to be provided, which is not possible in our case since there are only target prompt and original image given in our settings. SDEdit (Meng et al., 2021) utilizes DDIM Inversion to add noises to the original image and then denoises the image with target prompt. Blended Diffusion (Avrahami et al., 2022) requires a user provided mask and performs text guided image editing with SDEdit. DiffEdit (Couairon et al., 2023) obtains the target object mask with Diffusion Model itself by a user provided source prompt and conduct SDEdit in the mask region. PnP Diffusion (Tumanyan et al., 2023) injects intermediate features of original image to the generation of target prompt. Instruct pix2pix (Brooks et al., 2023) pretrains the Diffusion Models on external datasets with triplets of original image, edited image and target prompt. All these non-optimization methods suffer from the fact that they are either incapable of preserving the characteristics or unable to conduct complex non-rigid editing. Drag Diffusion (Mou et al., 2023), which extends DragGAN (Pan et al., 2023) to

Diffusion Models, is capable of preserving the identity of the original image and performs non-rigid edits. However, Drag Diffusion and DragGAN are only capable of performing space-related editing, which is just a portion of general image editing tasks. Instead, our Forgedit is a general text guided image editing framework to conduct various kinds of image editing operations, including spatial transformations. Prompt to Prompt (Hertz et al., 2023) requires that the source prompt and target prompt must be provided in a precisely matching form so that the algorithm could accurately find the editing target, which is too ideal thus impossible in our setting. Imagic (Kawar et al., 2023) is a three-stage optimization based editing method, which is the current state-of-the-art text guided image editing algorithm, which could be regarded as a combination of textual inversion (Gal et al., 2023)in the first stage and DreamBooth (Ruiz et al., 2023)in the second stage. However, the fine-tuning stages of Imagic are very slow and suffer from overfitting.

### A.4 HYPERPARAMETERS EFFECTS FOR TEXT EMBEDDING INTERPOLATION

We explore the effects of hyperparameters in vector subtraction and vector projection in Figure 9.

### A.5 WHAT SHOULD SOURCE PROMPT BE?

We explore the importance of using generated caption as source prompt. We use BLIP generated source prompt to describe the original image, yet previous SOTA method Imagic uses target prompt as source prompt. Since target prompt indicates the editing target, it is obviously inconsistent with the original image. Although target prompt embedding is optimized with the original image for reconstruction, it is carefully controlled by learning rate and made sure not far from the initial target prompt embedding. In many cases, using target prompt as source prompt confuses the editing model to correctly understand the editing intention of target prompt, making it lose the ability to conduct the edit, even after finetuning. In Figure 10, we show cases that do not need to use forgetting strategy from Figure 7, so that we could remove the effects of forgetting strategy. If target prompt is used instead of BLIP generated source prompt, all these cases of Forgedit without using generated source prompt will overfit.

### A.6 LIMITATIONS

First of all, although our fine-tuning framework has been optimized and is much faster than Imagic, the fine-tuning process still takes tens of seconds or even more depending on the GPU devices. We will explore in the future whether it is possible to preserve high fidelity characteristics of the original image without fine-tuning. Second, the effect of Forgedit is influenced by randomness. The fine-tuning process inevitably introduces randomness thus for some particular cases, we cannot guarantee to perfectly reconstruct the details of original image thus we have to run the fine-tuning stage several times for these challenging cases. The sampling procedure is also related to the initial random seed of reverse process, thus for some extremely challenging cases we have to sample tens of images or even hundreds, though rarely the case, before we could get a proper edited one. Third, the editing capability of Forgedit is restricted by the utilized Diffusion Model. If the target prompt cannot even be generated by the Diffusion Model itself, it is almost impossible to accomplish the edit according to the target prompt. For example, the prompt 'a sitting flamingo' cannot be generated by Stable Diffusion at all, thus Forgedit cannot successfully edit it either. Such an issue could possibly be solved by switching to better Diffusion Models.

*original image*

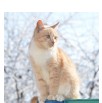

*target prompt*: *A photo of a cat wearing a hat.*

*vector subtradtion* $(1 - \gamma)e_{src} + \gamma e_{tgt}$

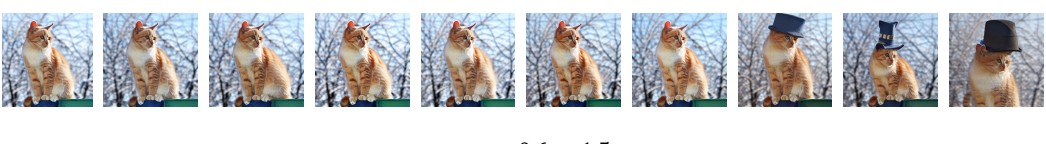

$\gamma = 0.6 \; to \; 1.5$

*vector projection* $\alpha e_{edit} + \beta e_{src}$

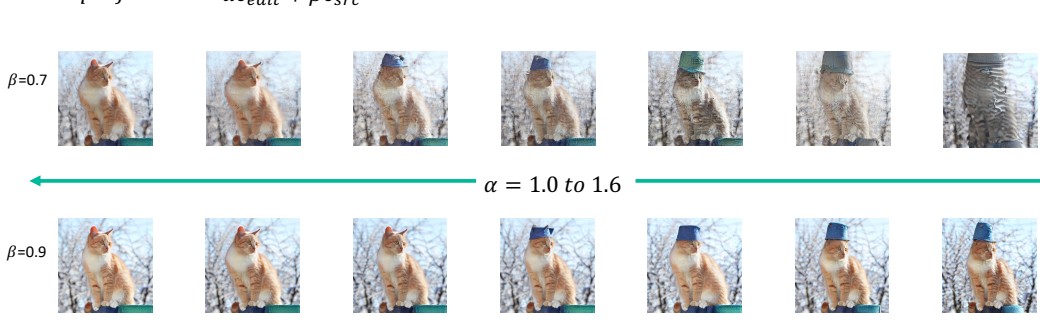

$\alpha = 1.0 \; to \; 1.6$

Figure 9: $\gamma$ for vector subtraction and $\alpha, \beta$ for vector projection.

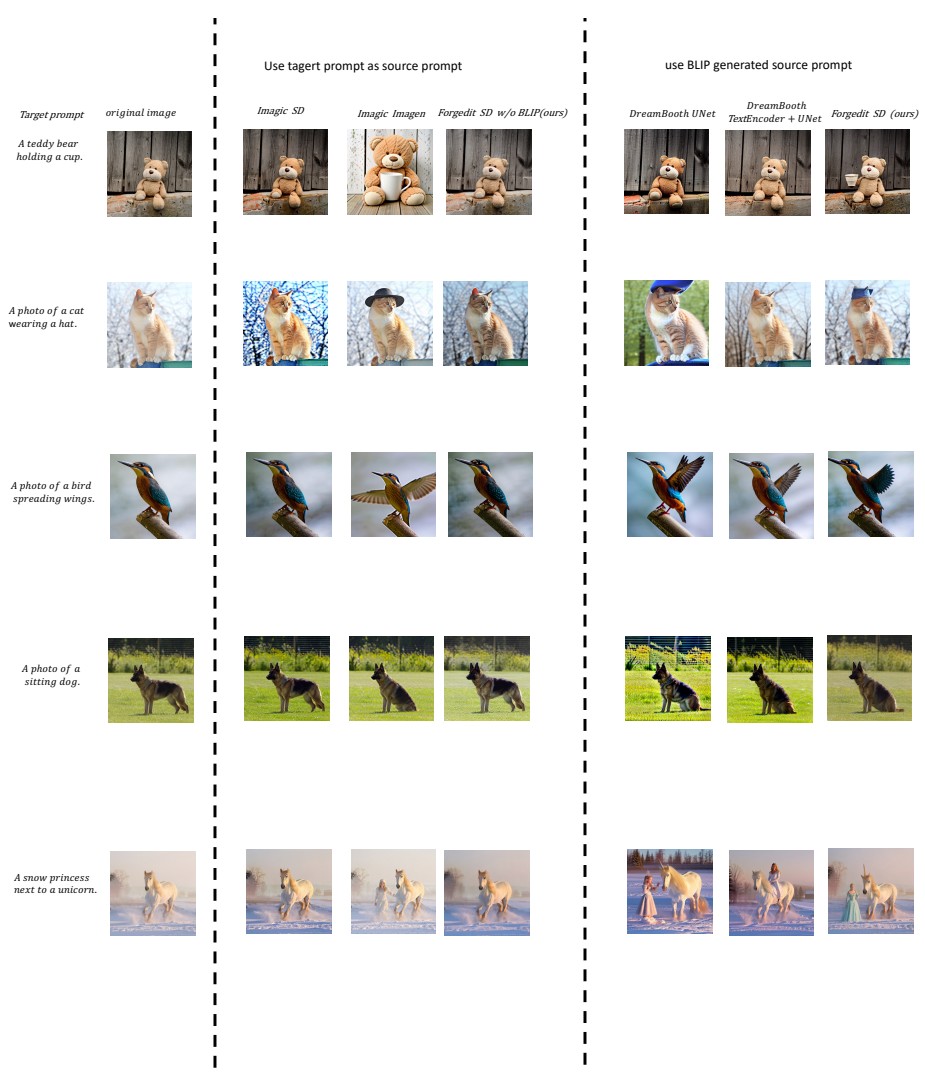

Figure 10: What should the source prompt be? We could find that Forgedit using target prompt leads to severe overfitting, yet Forgedit using BLIP generated source prompt eases overfitting.

