# OpenReview forum: "Forgedit: Text Guided Image Editing via Learning and Forgetting"
_ICLR.cc/2024/Conference — Submitted to ICLR 2024_

### Official Review · Reviewer_kPUJ · 2023-10-29

**Soundness:** 3 good
**Presentation:** 3 good
**Contribution:** 2 fair
**Rating:** 5
**Confidence:** 3

**Summary:**

This paper proposes a framework to tackle some issues like overfitting and inconsistency in common optimization-based editing methods. It presents a novel vector projection mechanism to merge the source text embeddings and the target embeddings to better preserve inversion consistency. Finally, it proposes the forgetting strategy during sampling to overcome the common overfitting issue. Empirically, it achieves SoTA performance in TEdBench.

**Strengths:**

– The paper has many interesting empirical observations, like using BLIP caption instead of the target prompt as the source text is more generalized and finetuning only the first few layers of the encoder/decoder leads to better generalization as well.

 – The proposed vector projection mechanism is intuitive and effective, and can better preserve the visual appearance from the ablation study.

**Weaknesses:**

– It is not clear why using BLIP caption as source text embedding can avoid overfitting, it is empirically observed but no explanation from the authors

– The proposed forgetting strategy is not general. Although the author ablates a lot of forgetting layers, it is not clear how to apply this strategy in the practical use case. I hope the author can provide a clear conclusion on how to choose the forgetting layers.

**Questions:**

– I have concerns about the optimization time required for embedding the real image, i.e. each edit operation becomes harder and longer at the wait time, what are your options on balancing editing quality and editing interactiveness?

---

> ### Author Response · Authors · 2023-11-23
>
> Thank you.
>
> Q1: It is not clear why using BLIP caption as source text embedding can avoid overfitting, it is empirically observed but no explanation from the authors
>
> A1: Since in previous SOTA method Imagic, target prompt is used as source prompt to describe the original image. Target prompt is the editing target, which is obviously inconsistent with the original image before editing. Although target prompt embedding is optimized with the original image for reconstruction, it is carefully controlled by learning rate and made sure not far from the initial target prompt embedding. In many cases, using target prompt as source prompt confuses the editing model to correctly understand the editing intention of target prompt, making it lose the ability to conduct the edit,  even after finetuning, which we show in the revised paper in Figure 10 in the appendix.
>
> Q2: The proposed forgetting strategy is not general. Although the author ablates a lot of forgetting layers, it is not clear how to apply this strategy in the practical use case. I hope the author can provide a clear conclusion on how to choose the forgetting layers.
>
> A2: The principles of the forgetting strategy is  based on a general property of UNet we found in Diffusion Models, i.e., encoder of UNet controls spatial information, for example, object pose, action and position, decoder of UNet controls appearance and identity. With such a finding, when facing overfitting issues, we could determine whether to forget encoder or decoder parameters according to the editing intention of target prompt.
>
> To be specific, we first run vector subtraction and vector projection. Only when the editing results are overfitting, i.e., unable to complete the edit, we apply the forgetting strategies. With the unet property we found, if the target prompt is related with spatial changes, we use forgetting strategies to unet encoder. If the target prompt is related with appearance or identity changes, we use forgetting strategies to unet decoder. In terms of the which forgetting strategy to use, here we take encoder for instance.  We start from forgetting everything in encoder except all attention modules, which we called 'encoderattn' in Figure 6. This is the default forgetting strategy which we found could tackle most overfitting cases in terms of space related target prompt. If we found that the editing results are still overfitting, we could forget more parameters with the strategy 'noencoder' in Figure 6, which means that we forget all encoder parameters. Instead, if we found that the editing can be complete yet the spatial structures change too much, we could forget less parameters with 'encoderattn+encoder1', which preserves all attention modules and block 1 in unet encoder. In the future, we would like to explore the possibility of incorporating visual LLM to make decisions automatically. Instead, if there is no human in the loop to choose forgetting strategies, we could simply ran all forgetting strategies during sampling and mannually select the best editing result.
>
>
>
>
> Q3: I have concerns about the optimization time required for embedding the real image, i.e. each edit operation becomes harder and longer at the wait time, what are your options on balancing editing quality and editing interactiveness?
>
> A3: Our Forgedit takes 40 seconds to optimize on each image compared with Imagic+SD taking 7 minutes. Of course it could be possible to decrease the optimization time by decreasing the training steps. However, the cost could be that the quality of reconstruction will also be influenced. Also it is possible to replace the optimization process with tuning-free methods like masactrl. However, the characteristics of original image may not be preserved well for such tuning-free methods.

---

### Official Review · Reviewer_xeZb · 2023-10-30

**Soundness:** 3 good
**Presentation:** 3 good
**Contribution:** 2 fair
**Rating:** 3
**Confidence:** 4

**Summary:**

The paper introduces Forgedit, a novel text-guided image editing method that addresses challenges in preserving image characteristics during complex non-rigid editing. It employs an efficient fine-tuning framework, vector subtraction, projection mechanisms, and innovative forgetting strategies inspired by UNet structures in Diffusion Models. Forgedit outperforms previous methods on the TEdBench benchmark, achieving state-of-the-art results in both CLIP and LPIPS scores for text-guided image editing.

**Strengths:**

This paper overall is clear and easy to follow.

**Weaknesses:**

1. Although, the paper has presented convincing results to solve image editing problems of diffusion model, the bag of tricks are now new and just work as expected.

2. Vector subtraction has been widely used in generative image editing, in VAEs, GANs and diffusion models.

3. Vector projection is a kind of component analysis, which has been well studied in latent code manipulation in GANs.

4. Using captioner to get source prompt is straightforward, and usually it's not even required, since vision-language learning is applied.

5. Many related editing works are missing, like plug-and-play, prompt-to-prompt, etc.

6. Model ensemble has been well-known to alleviate forgetting problems, both discriminative and generative modeling.

7. How does the hyper-parameters in vector subtraction and projection affect editing results, content and editing fidelity?

**Questions:**

See above in Weakness.

---

> ### Author Response · Authors · 2023-11-23
>
> Thank you.
>
> Q1: Although, the paper has presented convincing results to solve image editing problems of diffusion model, the bag of tricks are now new and just work as expected.
>
> A1: We respect reviewer's opinion and we still would like to briefly conclude our main novelties:
> 1. vision language joint optimization, much faster than previous SOTA and less overfitting.
> 2. we proposed vector projection mechanism to simultaneously control identity preservation and control editing strength.
> 3. we discovered a general unet property, i.e., unet encoder for space and structure, unet decoder for appearance and identity. We design forgetting strategies according to such unet property and target prompt.
>
> Q2: Vector subtraction has been widely used in generative image editing, in VAEs, GANs and diffusion models.
>
> A2: We never claim the vector subtraction is our novelty and we have mentioned that such a simple interpolation method was utilized in the Imagic. We totally agree with reviewer on this point.
>
> Q3: Vector projection is a kind of component analysis, which has been well studied in latent code manipulation in GANs.
>
> A3: vector projection mechanism is one of our contributions in this paper. According to our study,  we are the first to propose such a text embedding decomposition mechanism in Diffuison Models and the first to successfully demonstrate its functionality in text guided image editing problem.
> Such decomposition mechanism in text embedding space has never been explored, neither by computer vision community nor NLP community.
> Previous component analysis are applied to latent code manipulation in GANs. However, please note that there is no latent code for Diffusion Models at all.
>
>
> Q4: Using captioner to get source prompt is straightforward, and usually it's not even required, since vision-language learning is applied.
>
> A4: We completely agree with reviewer that Using captioner to get source prompt is straightforward. However, we would like to argue that it is vitally required. Using such an extra source prompt instead of target prompt is essential for preventing overfitting. We could find evidence to support this claim from two aspects:
>
>  1. Imagic uses target prompt as source prompt yet our Forgedit uses BLIP generated caption as source prompt. This straightforward modification leads to significant difference in the overfitting problem even WITHOUT using Forgetting strategies.  For example, in Figure 7, for the cases 'A photo of a teddy bear holding a cup.', 'A photo of a cat wearing a hat.', 'A photo of a bird spreading wings.', 'A photo of a sitting dog.','A snow princess next to a unicorn.', our editing results in the last column are using complete finetuned parameters without applying forgetting strategies. Please compare the Imagic+SD column and our last column for these cases, which demonstrate that using BLIP captions eases overfitting problems of Imagic.
>  2. We add new evidence Figure 10 in the appendix to further support our claim. In our Forgedit framework, we only replace the source prompt generated by BLIP to target prompt. The results demonstrate that using target prompt as source prompt still leads to severe overfitting issue, though joint vision-language learning is applied.

---

> ### Author Response · Authors · 2023-11-23
>
> Q5: Many related editing works are missing, like plug-and-play, prompt-to-prompt, etc.
>
> A5: They were not missing. They were mentioned in the relate work section and we discussed their advantages and disadvantages. Due to page limit of ICLR, we had no choices yet had to move the related work section to appendix, which might cause the reviewer to miss our mentioning of these papers.
>
> To be brief, plug-and-play does not work on non-rigid editing. Prompt-to-prompt requires a user input source prompt which precisely matches the target prompt in a word-by-word manner, which is impossible in our text guided image editing setting where only target prompt and original image are provided.
>
>
> Q6: Model ensemble has been well-known to alleviate forgetting problems, both discriminative and generative modeling.
>
> A6:
> 1.  Our forgetting mechanism could be regarded as model ensembles yet we utilize such forgetting mechanism to alleviate the overfitting problem, not forgetting problems.
>
>  2.  Our main contribution of the forgetting strategy is that we found a general property of UNet in Diffusion Models, i.e., encoder of UNet controls spatial information, for example, object pose, action and position, decoder of UNet controls appearance and identity. With such a finding, when facing overfitting issues, we could determine whether to forget encoder or decoder parameters according to the editing intention of target prompt.
>
>
> Q7: How does the hyper-parameters in vector subtraction and projection affect editing results, content and editing fidelity?
>
> A7: We add the exploration of these hyper parameters in Figure 9 in the appendix.

---

### Official Review · Reviewer_9c4b · 2023-10-31

**Soundness:** 3 good
**Presentation:** 3 good
**Contribution:** 3 good
**Rating:** 6
**Confidence:** 2

**Summary:**

The paper proposes the Forgedit for text-guided image editing. There are three key components in the Forgedit:1) Fine-tuning the framework and the text embeddings jointly; 2) The vector subtraction and projection for image editing; 3) Forgetting strategy in the UNet-Structures. The proposed method achieves state-of-the-art performance.

**Strengths:**

1) The paper performs extensive explorations on diffusion-based image editing. The mechanisms the authors explore include the difference between vector subtraction and projection, changes brought by keeping and dropping different weights of unet. These explorations are meaningful and can provide insights to readers.

2) The paper is well-organized and easy to follow.

3) The proposed method achieves state-of-the-art performance on the image editing benchmark.

**Weaknesses:**

1) There are many components that should be adjusted at the inference time. It is troublesome to adjust all these parameters manually.

2) For vector subtraction and vector projection, we need to decide which variant to use and also there are some hyper-parameters in these two variants that need to be determined.

3) For Fig. 5 and Fig. 6, it is hard to tell the settings of each column from the captions.

4) In Table 1, the quantitative results of other methods are missing.

**Questions:**

Please see my concerns in the weakness part.

---

> ### Author Response · Authors · 2023-11-23
>
> Thank you.
>
> Q1: There are many components that should be adjusted at the inference time. It is troublesome to adjust all these parameters manually.
>
> A1:
> TLDR: We first inference without forgetting strategies. If overfitting happens, we choose from the default 'encoderattn' or 'decoderattn' strategy according to the UNet property and target prompt intention.
>
> In fact, we ran the TEdBench benchmark with only three forgetting strategies, i.e., forgetting everything in encoder except all attention modules, forgetting everything in the decoder except attenton modules, and forget nothing. This is because we found that these three strategies are empirically rather general and could tackle most overfitting issues. This means that for automatic inference, we only need to infer four loops, which are vector subtraction with three forgetting strategies, and vector projection.
>
> If there are human to interact, one could first run 2 loops with vector projection and vector subtraction without forgetting. If the editing is unsuccessful due to overfitting, one could determine to apply forgetting strategies in encoder or decoder. This is due to our finding of the general property of UNet for its different functionalities in terms of encoder and decoder. If the target prompt aims to edit the appearance, for example, turn a zebra to a horse, then forgetting strategy should be applied to decoder. If the target prompt aims to edit pose, action or image layout, for example, let the horse raise it head, one should apply forgetting strategy in encoder.
>
> In terms of which forgetting strategy to choose, forgetting everything except all attention modules is generally a good starting point. The user could adjust to utilize other forgetting strategies to forget more or less according to the editing results.
>
> Generally, such a process leads to at most 4 loops before successful edit. We choose to offer these choices in order to guarantee success rate. With human in the loop to make choices, the editing process should be very clear and straight forward. We are also considering introducing visual LLM in the future to automatically understand the editing intention from input image and target prompt, and automatically make editing choices.

---

> ### Author Response · Authors · 2023-11-23
>
> Q2: For vector subtraction and vector projection, we need to decide which variant to use and also there are some hyper-parameters in these two variants that need to be determined.
>
> A2: For automatic running our algorithm on TEdBench, we ran both vector subtraction and projection, whose hyper-paramters ranges are introduced in the 'editing' in section 2.3, where we iterate over these ranges to sample around 10 images. The final editing results are selected from these images, either manually or automatically according to metrics should be fine.
>
> Q3:For Fig. 5 and Fig. 6, it is hard to tell the settings of each column from the captions.
>
> A3: We elaborated the detailed settings  of these colums in ablation study section 3.1.
>
> Q4: In Table 1, the quantitative results of other methods are missing.
>
> A4: In fact, since most image editing papers do not compare on the same benchmark, Imagic+Imagen is the only method providing their editing results on TEdBench.  We asked some authors of other methods for their quantitative results on TEdBench yet we got no reply. We did run some other methods on this benchmark by ourselves and their results on this benchmark are bad. However, we doubt whether it is appropriate and convincing to be us who claim these results of other methods on TEdBench instead of the authors themselves.

---

### Official Review · Reviewer_MyDZ · 2023-11-04

**Soundness:** 3 good
**Presentation:** 3 good
**Contribution:** 3 good
**Rating:** 5
**Confidence:** 4

**Summary:**

The paper introduces an optimization-based image editing method capable of performing both rigid and non-rigid editing. Additionally, the paper proposes a forgetting strategy within the UNet architecture of diffusion models to prevent overfitting. Experimental results demonstrate the effectiveness of the proposed method.

**Strengths:**

1. The writing is clear and easy to follow.
2. To achieve the desired editing, the authors propose an adaptation of DreamBooth and also incorporate the optimization strategy from Imagic. To address potential overfitting arising from a single input image, a forgetting strategy is introduced.
3. The experiments provide evidence of the effectiveness of the proposed method, both in the context of rigid and non-rigid editing.

**Weaknesses:**

1. The training strategy of the proposed method is similar to Imagic, with the main differences being that the authors employ BLIP to generate a caption describing the input image, and combine the first and second stages in Imagic into one. Besides, authors use DreamBooth as the backbone.

2. I find the location of the point (1-y)e_src + ye_tgt in Figure 2 confusing, and I'm uncertain why the value of y (gamma) exceeds 1 in vector subtraction. Typically, y should fall within the range [0,1] if normalization has been applied. Furthermore, It would be beneficial to include a discussion explaining why projection is more suitable for editing compared to vector subtraction, in terms of identity preservation.

3. The qualitative comparison suggests that the results produced by the proposed method may have lower resolution compared to other methods, as evident in the examples of the dog, bird, and giraffe in Figure 7. I am concerned about the potential impact of the proposed method on image quality, and I notice that there are no evaluation metrics in the paper reflecting image quality, such as Inception Score (IS) and Fréchet Inception Distance (FID).

4. It would be beneficial to include a quantitative comparison for the various components employed in the proposed method. Additionally, it's unclear why the authors chose to apply the forgetting strategy only in vector subtraction and not in projection. Further clarification on this decision would be helpful.

**Questions:**

Please see above weaknesses.

---

> ### Author Response · Authors · 2023-11-23
>
> Thank you.
>
> Q1: The training strategy of the proposed method is similar to Imagic, with the main differences being that the authors employ BLIP to generate a caption describing the input image, and combine the first and second stages in Imagic into one. Besides, authors use DreamBooth as the backbone.
>
> A1: Yes,the finetuning stage of our method could be regarded as a variant of Imagic by combining the first stage and the second stage into one and uses BLIP generated caption as source prompt instead of using target prompt as source prompt like what Imagic does. Such modification may seem like limited novelty. However, we found that
> 1. First, our vision language joint learning is the key to much faster convergence speed than Imagic and leads to much fewer training steps. With our joint learning of image and text embedding, the finetuning stage of one image with our Forgedit+Stable Diffusion takes 40 seconds on an A100 GPU, compared with 7 minutes with Imagic+Stable Diffusion reported by Imagic paper. This leads to more than 10 times speed up.
> 2. Second, using BLIP generated captions eases overfitting in many cases. Although we introduce Forgetting strategies to tackle the overfitting problem in image editing, using BLIP generated captions instead of target prompt as source prompt also ease the overfitting problem. For example, in Figure 7, for the cases 'A photo of a teddy bear holding a cup.', 'A photo of a cat wearing a hat.', 'A photo of a bird spreading wings.', 'A photo of a sitting dog.',' A snow princess next to a unicorn.', our editing results in the last column are using complete finetuned parameters without applying forgetting strategies. Please compare the Imagic+SD column and our last column for these cases, which demonstrate that using BLIP captions eases overfitting problems of Imagic. We also further experiment with our Forgedit by replacing the BLIP generated source prompt to target prompt and observed severe overfitting issues in Figure 10 in the revised paper in the appendix. Please check it.
> 3. We use DreamBooth as an alternative finetuning method in order to demonstrate that our Forgedit is a general framework compatible with other finetuning methods.
>
>
> Q2: I find the location of the point (1-y)e_src + ye_tgt in Figure 2 confusing, and I'm uncertain why the value of y (gamma) exceeds 1 in vector subtraction. Typically, y should fall within the range [0,1] if normalization has been applied. Furthermore, It would be beneficial to include a discussion explaining why projection is more suitable for editing compared to vector subtraction, in terms of identity preservation.
>
> A2:
> 1. For the value of gamma exceeding 1, we observe such phenomenon in many cases of TEdBench. Empirically, although for some cases the  gamma value falls in range [0,1], for the others editings are tough and  we need to increase the editing strength to complete the edit and gamma goes beyond 1.
> 2. Vector projection decomposes the target embedding e_tgt to a subvector along source embedding e_src and a subvector e_edit orthogonal to source embedding. We use two different hyperparamters to control the strength of the source embdding e_src and editing embedding e_edit. Considering the fact that there is only one hyperparamter gamma in vector subtraction, vector projection can seperately control the similarity to original image and strength of editing, which is why vector projection is better at identity preservation.

---

> ### Author Response · Authors · 2023-11-23
>
> Q3: The qualitative comparison suggests that the results produced by the proposed method may have lower resolution compared to other methods, as evident in the examples of the dog, bird, and giraffe in Figure 7. I am concerned about the potential impact of the proposed method on image quality, and I notice that there are no evaluation metrics in the paper reflecting image quality, such as Inception Score (IS) and Fréchet Inception Distance (FID).
>
> A3: In fact, all images except the original image column and the Imagic+Imagen column are all of resolution 512x512. So our understanding is that reviewer indicates that the resolution of Imagic+Imagen is higher than other images. This is due to  the fact that we use Stable Diffusion 1.4 as base model, whose output is of resolutioin 512x512, and Imagic+Imagen uses Imagen as base model thus outputs images of 1024x1024 resolution which may be even higher than  original image's. In order to increase the resolution, we could use on-the-shelf super resolution models,  for example, stable diffuison upscalex4 model to increase the resolution from 512x512 to 1024x1024. We have updated the FID score on TEdBench in Table 1, where our Forgedit achieves better FID score than Imagic. It is also possible to adapt our Forgedit to SDXL to directly generate 1024x1024 images, which we will explore in the future.
>
> Q4: It would be beneficial to include a quantitative comparison for the various components employed in the proposed method. Additionally, it's unclear why the authors chose to apply the forgetting strategy only in vector subtraction and not in projection. Further clarification on this decision would be helpful.
>
> A4: Thanks for the advice. We will include the quantitative experiments for seperated components in the future version, since the reviewing period is rather limited and extra quantitative experiments takes quite a lot of workloads on TEdBench benchmark. We provide more qualitive ablation studies in the appendix.
>
> The reason we chose to apply forgetting strategies on vector subtraction instead of vector projection is simply that we do not want to introduce  too much extra sampling time  since we ran the benchmark with automatic algorithms. It is of course alright to also apply the forgetting strategies to vector projection yet we found that generally apply forgetting strategies to vector subtraction should be enough to tackle the overfitting issues well.

---

### Author Response · Authors · 2023-11-23

We thank all reviewers for their valuable comments. We have revised the paper to stress reviewers' concerns.
1. We use super resolution models to upscale from 512x512 to 1024x1024 shown in Figure 7 and calculate FID score in Table 1, where our Forgedit+Stable Diffusion 1.4 achieves better FID score than previous SOTA Imagic+Imagen.
2. We demonstrate the effect of hyperparemeters in vector projection and vector subtraction in Figure 9 in the appendix.
3. We demonstrate that using target prompt as pseduo source prompt leads to overfitting and could be tackled by using BLIP generate source prompt in Figure 10.

We also would like to briefly conclude our main contributions:
1. A vision language joint optimization framework for text guided image editing, much faster than previous SOTA and much less overfitting.
2. We propose vector projection mechanism in text embedding space of Diffusion Models, which is capable to control the identity and editing strength seperately.
3. We discovered a general property of UNet in Diffusion Models, i.e., Unet encoder learns space and structure, Unet decoder learns appearance and identity. With such a property, we design a forgetting mechanism to successfully tackle the vital overfitting issues of image editing.

---

### Meta-Review · Area_Chair_trpw · 2023-12-05

**Metareview:**

This paper proposes Forgedit for image editing. The key component of the method is the forgetting mechanism to avoid overfitting. This paper receives mixed ratings of (3, 5, 5, 6) and no consensus is reached. The reviewers appreciate the convincing results obtained by Forgedit, but have concerns on the technical contributions of the method. In particular, the proposed method is similar to Imagic. After reading the paper and rebuttal, the AC shares the concern about the novelty, and therefore a rejection is recommended.

**Justification For Why Not Higher Score:**

While the proposed method demonstrates convincing performance, the AC shares the concern of reviewers about the technical novelty of the method. Therefore, a rejection is recommended.

**Justification For Why Not Lower Score:**

N/A

---

### Decision · Program_Chairs · 2024-01-16

Reject